# Electrophysiological Correlates of Vocal Emotional Processing in Musicians and Non-Musicians

**DOI:** 10.3390/brainsci13111563

**Published:** 2023-11-07

**Authors:** Christine Nussbaum, Annett Schirmer, Stefan R. Schweinberger

**Affiliations:** 1Department for General Psychology and Cognitive Neuroscience, Friedrich Schiller University, 07743 Jena, Germany; annett.schirmer@uibk.ac.at; 2Voice Research Unit, Friedrich Schiller University, 07743 Jena, Germany; 3Institute of Psychology, University of Innsbruck, 6020 Innsbruck, Austria; 4Swiss Center for Affective Sciences, University of Geneva, 1202 Geneva, Switzerland

**Keywords:** vocal emotion perception, musicality, fundamental frequency (F0), timbre, parameter-specific voice morphing, electrophysiological correlates

## Abstract

Musicians outperform non-musicians in vocal emotion recognition, but the underlying mechanisms are still debated. Behavioral measures highlight the importance of auditory sensitivity towards emotional voice cues. However, it remains unclear whether and how this group difference is reflected at the brain level. Here, we compared event-related potentials (ERPs) to acoustically manipulated voices between musicians (*n* = 39) and non-musicians (*n* = 39). We used parameter-specific voice morphing to create and present vocal stimuli that conveyed happiness, fear, pleasure, or sadness, either in all acoustic cues or selectively in either pitch contour (F0) or timbre. Although the fronto-central P200 (150–250 ms) and N400 (300–500 ms) components were modulated by pitch and timbre, differences between musicians and non-musicians appeared only for a centro-parietal late positive potential (500–1000 ms). Thus, this study does not support an early auditory specialization in musicians but suggests instead that musicality affects the manner in which listeners use acoustic voice cues during later, controlled aspects of emotion evaluation.

## 1. Introduction

Listeners can infer a speaker’s emotional state from voice acoustics [1,2,3]. These acoustic cues include emotion-related modulations of fundamental frequency (F0) contour, timbre, amplitude, or timing. F0 refers to vocal fold vibrations and is perceived as pitch or—when it varies over time—pitch contour. Timbre may be understood as voice quality independent of pitch, timing, and volume and enables listeners to distinguish, for example, a “harsh” from a “gentle” voice. Amplitude refers to an utterance’s perceived loudness, while timing refers to its temporal characteristics, such as duration, speech rate, and pauses. The emotional state of a speaker affects virtually all of these acoustic features [1]. Hot anger, for example, is often characterized by a high F0, a rough timbre, a large amplitude, and a fast speech rate, whereas the opposite pattern is observed for sadness.

As a consequence, emotion perception involves the constant monitoring and integration of relevant acoustic cues in real time. This capacity builds on a complex and rapid neural network [4,5]. Many studies using electroencephalography (EEG) showed that vocal emotional processing in the brain follows a distinct time course comprised of several sub-processes, which can be linked to different event-related potential (ERP) components [5,6]. An initial and largely bottom–up driven analysis of voice acoustic cues has been linked to the N100 component [7,8]. This is followed by cue integration to derive emotional meaning, which unfolds as early as 200 ms past voice onset and can modulate components like the P200 and the mismatch-negativity (MMN) [9,10,11,12,13,14]. Subsequent processes involve more top–down-regulated and goal-directed analyses, such as the integration of vocal with verbal content, explicit emotion evaluation, or response preparation, which are associated with components like the N400 or the late positive potential (LPP) [10,12,13].

All the above ERP components were found to be sensitive to changes in acoustic parameters such as pitch, loudness, and timbre [7,12,13]. However, our understanding of how specific acoustic cues are processed to derive the emotional quality of voices is still incomplete [5]. A key problem concerns the natural co-variation and redundancy of acoustic cues in emotional voices, making it difficult to study their isolated contributions [2]. Modern voice manipulation techniques offer experimental control over the acoustic properties of vocal emotions and thus permit addressing this problem [15]. One of these techniques is parameter-specific voice morphing. Here, original voice recordings may be manipulated by altering some or all of their acoustic parameters. For example, vocal stimuli can be resynthesized such that they express emotional information through only one acoustic cue (e.g., F0) while rendering other cues (e.g., timbre) uninformative. We recently established this approach in the context of an EEG study on vocal emotion perception [13]. Specifically, we created voices expressing happiness, sadness, pleasure, and fear through F0 or timbre only, while the other parameters were held at an emotionally non-informative level. We found both reliable ERP modulations (P200 and N400) and behavioral effects of these acoustic manipulations, which suggested that F0 was relatively more relevant for the processing of happy, fearful, and sad voices, whereas timbre was relatively more important for the processing of voices expressing pleasure. Intriguingly, the N400-amplitude difference in response to F0 vs. timbre predicted the behavioral recognition difference between F0 and timbre stimuli, suggesting that the acoustic manipulation affected not only sensory processes but also neural mechanisms linked to evaluative judgments.

One important observation is that the processing of emotional voice cues, indexed by both behavioral and electrophysiological responses, is subject to profound individual differences [5,14,16,17]. A particularly interesting trait in this context is musicality. At the behavioral level, there is consistent evidence that musicians outperform non-musicians in vocal emotion recognition [18,19,20,21]. Crucially, empirical findings converge in highlighting the importance of auditory sensitivity towards emotional voice cues as a key mechanism underlying this performance difference [18,22,23,24,25]. Compared to non-musicians, musicians have been found to be more sensitive to pitch, timbre, loudness, and temporal aspects of sounds [26,27]. In the specific context of vocal emotions, sensitivity to the pitch contour (F0) seems to be of central importance [22,28,29]. In a recent study [29], we showed that musicians outperformed non-musicians when emotions were expressed by F0 only and by the combined modulation of F0 and timbre, but not when emotions were expressed through timbre only.

While these behavioral findings suggest that musicians are particularly tuned to emotional information expressed through pitch cues (F0), it remains unclear whether and how this group difference is reflected at the brain level. Effects of musicality can be observed in electrophysiological brain responses to auditory stimuli [26,30,31,32,33,34,35]. For example, musicality has been found to modulate the N100, the P200, and the mismatch negativity (MMN) in response to musical stimuli [30,31,36], although evidence for the N100 is somewhat inconclusive [37,38]. Beyond musical stimuli, differences between musicians and non-musicians have been reported for speech [39,40,41] and nonverbal vocal expressions [35]. However, findings that targeted vocal emotion perception are sparse and inconsistent [42,43,44]. One study compared ERP responses to music and nonverbal vocalizations in an implicit listening task, where emotional voices were presented while participants had to detect intermediate non-vocal sounds [45]. The authors reported differences between musicians and non-musicians in terms of larger amplitudes in musicians at central and fronto-central electrodes in the P200, the P300, and the LPP, but only in response to musical stimuli and not in response to emotional vocalizations. To which degree these findings generalize to explicit listening tasks, and emotional prosody perception remains unresolved. To the best of our knowledge, no study has compared ERP responses of musicians and non-musicians to acoustically manipulated emotions to explore how individual differences in sensitivity to emotional F0 and timbre cues are reflected at the brain level.

The aim of the present study was to assess differences in the ERP response to vocal emotions in musicians and non-musicians while at the same time conceptually replicating the parameter-specific ERP modulations, i.e., the effects of the F0 and timbre manipulation, observed in our previous study [13]. To this end, we re-invited the participants of the behavioral study reported in [29] to the lab and recorded their EEG, adopting a similar protocol as in [13]. If the behavioral benefits are due to early, automatic representations of parameter information, then musicians and non-musicians should differ in the P200. The former group may show more pronounced parameter effects than the latter. If, however, behavioral benefits arise at later more controlled aspects of voice processing, we might see them in the N400 or the LPP, which is larger in response to stimuli of higher emotional significance [10,12]. Amplitude differences in these components may reveal that parameter representations inform emotion-evaluative processes in musicians and non-musicians differently.

## 2. Materials and Methods

### 2.1. Participants

We recorded the EEG from 80 participants in total between June 2021 and May 2022. Participants were recruited through mailing lists at the Friedrich Schiller University Jena, the University of Music Franz Liszt Weimar, as well as several music schools and orchestras in Jena. Advertisements for musicians and non-musicians were distributed separately, stating the different inclusion criteria. Before coming to the EEG lab, all participants completed a behavioral online study (more details in the procedure below and [29]). Thus, with the exception of very few excluded datasets, samples of the behavioral study reported in [29] and the current EEG study are identical. Two participants (one musician and one non-musician) had to be excluded due to poor EEG data quality (extensive drifts and muscle artifacts in both cases), resulting in 78 final datasets. All participants were aged between 18 and 50 years and fluent in German. Informed consent was provided before completing the experiment, and data collection was pseudonymized. Participants were compensated with 25 € or with course credit. The experiment was in line with the ethical guidelines of the German Society of Psychology (DGPs) and approved by the local ethics committee (Reg.-Nr. FSV 19/045).

#### 2.1.1. Musicians

Data from 39 (semi-) professional musicians entered analysis (18 males, 21 females, aged 20 to 42 years; *M* = 29.9, *SD* = 5.48). Mean onset age of musical training was 7 years (*SD* = 2.54, 4–17 years). Twenty-three participants were professional musicians with a music-related academic degree; all others had a non-academic music qualification (i.e., they worked as musicians or won a music competition). Thirty-five participants had studied their instrument for over 10 years, three between 6 and 9 years, and one between 4 and 5 years.

#### 2.1.2. Non-Musicians

Data from 39 non-musicians entered analysis (19 males and 20 females, aged 19 to 48 years; *M* = 30.5, *SD* = 6.34). Although our recruitment criteria specified that non-musicians had not learned an instrument and did not engage in any musical activities like choir singing during childhood, a couple of participants later reported some musical background (two reported 2 years and two reported 4–5 years of formal musical training; mean age at onset was 17 (*SD* = 10.52, range = 6–30 years); for details see our repository on the open science framework (OSF), Appendix A). Musical experience in these cases mostly encompassed mandatory flute lessons in primary school that were several decades ago. One participant was a hunter who used a special horn for this activity but played one pitch only. After consideration of each individual case, all participants were retained for data analysis.

### 2.2. Stimuli

#### 2.2.1. Original Audio Recordings

We selected original audio recordings from a database of vocal actor portrayals provided by Sascha Frühholz, similar to the ones used in [46]. Eight speakers (four male, four female) uttered three pseudowords (/molen/, /loman/, /belam/) with expressions of happiness, pleasure, fear, and sadness, resulting in a total of 96 recordings.

#### 2.2.2. Parameter-Specific Voice Morphing

In the first step, we created emotional averages using the Tandem-STRAIGHT software [47,48]. This was carried out via a weighted interpolation of all four emotions used in the study for each speaker and pseudoword (see Figure 1). The rationale behind the emotional averages was that although they are not neutral, they would be uninformative and unbiased with respect to the four emotions of interest. Thus, in the following, we used them as emotionally non-informative reference stimuli. In contrast to [13], where we used neutral voices as reference, we opted for emotional averages because a previous study showed that averages are more suitable for the subsequent generation of parameter-specific voice morphs, as they sounded more natural [49].

In the second step, we generated three parameter-specific morph types (see Figure 2). Full-Morphs were stimuli with all parameters taken from the emotional version (corresponding to 100% from the specific emotion and 0% from average), with the exception of the timing parameter, which was taken from the emotional average (corresponding to 0% specific emotion and 100% emotional average). F0-Morphs were stimuli with the F0-contour taken from the specific emotion, but timbre and timing taken from the emotional average. Conversely, Timbre-Morphs were stimuli with all timbre parameters (i.e., formant frequency, spectral level, and aperiodicity) taken from the specific emotion, but F0 and timing from the emotional average. Please note thatspectrum level refers to a representation of the spectral envelope, and aperiodicity refers to a representation of aperiodic sound components (more details in [48]). We also included the emotional averages for exploratory purposes. In total, this resulted in 8 (speakers) × 3 (pseudowords) × 4 (emotions) × 3 (morphing conditions) + 24 average (8 speakers × 3 pseudowords) = 312 stimuli. Using PRAAT [50], we normalized all stimuli to a root mean square of 70 dB SPL (duration M = 780 ms, range 620–967 ms, *SD* = 98 ms). For a more detailed description of the morphing procedure, see [29]. Stimulus examples and an overview of acoustic parameters can be found on OSF, Appendix A. Note that the stimulus material is identical to that used in [29].

### 2.3. Design

#### 2.3.1. EEG Setup

The EEG was recorded using a 64-channel BioSemi Active II system (BioSemi, Amsterdam, The Netherlands) with electrodes being attached to a cap in accordance with the extended 10–20 system (electrode specifications as in [51]). This system uses a common mode sense/driven right leg circuit instead of ground and reference electrodes (for further information, see https://www.biosemi.com/faq/cms&drl.htm (accessed on 25 September 2023)). The horizontal electrooculogram (EOG) was recorded from two electrodes at the outer canthi of both eyes, and the vertical EOG was monitored with a pair of electrodes attached above and below the right eye. All signals were recorded with direct current (120 Hz low-pass filter) and sampled at a rate of 512 Hz. During the EEG recording, participants were seated in a dimly lit, electrically shielded, and sound-attenuating cabin (400-A-CT-Special, Industrial Acoustics™, Niederkrüchten, Germany) with their heads on a chin rest to ensure a constant distance of 90 cm to the computer screen. The sound stimuli were presented via in-ear headphones (Bose^®^MIE2 mobile headset, Framingham, MA, USA). For the presentation of the written instructions and the stimuli, we used E-Prime 3.0 [52].

#### 2.3.2. Procedure

In a prior online study [29], participants entered demographic information, completed several questionnaires on personality traits and their musical background [53,54,55,56,57,58], and performed a music perception test [59,60]. Further, they performed an emotion classification experiment, which we did not analyze here, as it is reported in [29].

For the present lab study, participants were instructed to listen to the presented voices and pay attention to the vocally expressed emotions. Each trial started with a white fixation cross centered on a black screen. After 1000 ± 100 ms (jittered randomly), the cross changed into green, and a vocal stimulus started playing, followed by 2000 ms of silence, during which the green fixation cross stayed on the screen. In 10% of the trials, a prompting screen displaying the four response options “happiness”, “pleasure”, “fear”, and “sadness” appeared after voice offset, which lasted until the participant entered a response. This response prompt was included to ensure participants’ attentiveness towards the expressed emotions while at the same time reducing potential confounds related to the motor responses. The prompt was fully random, so the number of response trials differed across participants and conditions. Participants responded with their left and right index and middle fingers. The mapping of response keys to emotion categories was randomly assigned for each participant out of four possible key mappings and was identical for the EEG and the previous online session [29]. Emotions of the same valence were always assigned to one hand, and emotions with similar high vs. low intensities (fear–happiness vs. sadness–pleasure) were always assigned to the corresponding fingers of both hands (details on OSF, Appendix A). The experiment started with 20 practice trials with stimuli not used thereafter. Subsequently, all 312 experimental stimuli were presented once in random order and then again in a different random order, resulting in 624 trials. Individual self-paced breaks were allowed between blocks of 78 trials. The duration of the experiment was about 50 to 60 min.

### 2.4. Data Processing

The participants’ demographic information (e.g., age, sex, educational background), their personality and musical background questionnaires, as well as their music perception performance, were retrieved from the prior online session. We used these data for a detailed description of our sample and a comparison of musicians and non-musicians. Data from the emotion classification experiment were not analyzed here but are reported in [29].

EEG data were pre-processed using EEGLAB [61] in MATLAB R2020a [62]. Raw EEG recordings were downsampled to 256 Hz and re-referenced to the electrode average. Then, the data were low-pass filtered at 30 Hz, high-pass filtered at 0.1 Hz (both filters −6 dB/octave, zero-phase shift), and epoched using a time interval of −200 to 1000 ms relative to voice onset. Epochs were then visually scanned for noisy channels and other unsystematic artifacts, such as drifts or muscle movements. In one recording (musician), about 70 trials were lost due to malfunctioning headphones. In another one (non-musician), around 30 trials were lost due to extensive coughing of the participant. After visual inspection, ERPs of both datasets were found to be of sufficient quality and were therefore kept for analysis. Then, the data were 1 Hz high-pass filtered and subjected to an independent component analysis (ICA). The resulting component structure was applied to the pre-processed data with the 30 to 0.1 Hz filter settings. Components reflecting typical artifacts (e.g., eye movements) were removed before back-projecting information from component space into EEG channel space. Next, the data were baseline-corrected with a window of −200 to 0 ms relative to stimulus onset, and channels that had been removed earlier due to noise were interpolated using a spherical spline procedure [63] (one channel in 19 participants, two channels in 12 participants, and three channels in 3 participants). The resulting data were again scanned visually, and residual artifacts were removed. Remaining epochs were submitted to a current source density (CSD) transformation using the CSD toolbox in EEGLAB [64]. This transformation yields essentially reference-free data, which optimizes the segregation of spatially overlapping sources [65]. ERPs were derived by averaging epochs for each condition and participant. In total, a minimum of 32 trials and an average of 46.2 trials per condition (out of a possible maximum of 48) and participants entered statistical analysis. The condition with averaged emotions was excluded from analysis.

Based on previous findings [13], we focused on a fronto-central cluster (F1, Fz, F2, FC1, FCz, FC2, C1, Cz, C2), where we quantified mean amplitudes of the P200 (150 to 250 ms) and the N400 (300 to 500 ms). We opted for a slightly longer interval of the N400 than the one reported in [13], based on visual inspection of the averaged ERP waveform in the present data, because the N400 seemed to peak around 450 ms (details in the results section). In addition, we analyzed a later interval ranging from 500 to 1000 ms based on previous literature and visual inspection [66], which we refer to as late positive potential (LPP) (500 to 1000 ms). The LPP has a centro-parietal distribution [66] and was thus analyzed using a centro-parietal cluster (C1, Cz, C2, CP1, CPz, CP2, P1, Pz, P2).

### 2.5. Statistical Analysis

Statistical analysis and data visualization was performed in R Version 4.1.0 [67]. Comparison of musicians and non-musicians regarding demography, personality, and musicality was carried out using Chi^2^ and *t*-tests. For ERP analyses, we calculated ANOVAs with the mean amplitudes of the P200, the N400, and the LPP as dependent variables and the factors group, emotion, and morph type as independent variables. All reported intervals around effect sizes represent 95% confidence intervals.

Appendix A, analysis scripts, and pre-processed data can be found on the associated repository on the open science framework (OSF) https://osf.io/2jt5h/ (made public on 27 September 2023).

## 3. Results

### 3.1. Comparability of Musicians and Non-Musicians: Demography, Personality, and Musicality

Our samples of musicians and non-musicians were comparable with regard to socioeconomic status variables such as income (*χ*^2^ = 6.33, *df* = 4, *p* = 0.176), education (*χ*^2^ = 5.21, *df* = 2, *p* = 0.077), and highest professional degree (*χ*2 = 5.68, *df* = 8, *p* = 0.683), as well as age (|*t*(74.47)| = 0.46, *p* = 0.645), positive affective state (|*t*(75.42)| = 1.99, *p* = 0.051), negative affective state (|*t*(66.06)| = 1.39, *p* = 0.168), and most Big Five personality traits (all |*t*s| ≤ 1.54, *p*s ≥ 0.127), with one exception that musicians scored higher on Openness than non-musicians (|*t*(62.63)| = 2.55, *p* = 0.013). Further, while groups did not differ on overall autistic traits (|*t*(71.23)| = 1.60, *p* = 0.115), musicians scored higher on the Attention to Detail subscale (|*t*(75.96)| = 2.39, *p* = 0.019) and lower on the Social Communication subscale (|*t*(68.90)| = 2.47, *p* = 0.016). With regard to self-rated musicality as well as music perception performance, musicians scored considerably higher than non-musicians (all |*t*s| ≥ 3.03, *p*s ≤ 0.003, details on OSF, Appendix A).

### 3.2. Behavioral Data

On average, emotion classifications were prompted on 10% (*SD* = 1.3%, range = 7–13%) of the trials. The average proportion of correct classifications was M = 61% (*SD* = 8%), ranging from 38% to 81%, and was thus above the chance level of 25%. A visualization of results for each emotion and morph type is provided on OSF, Appendix A.

### 3.3. ERP

Mean amplitudes of the P200, the N400, and the LPP were analyzed in three different 2 × 4 × 3 ANOVAs with the between-subject factor group (musicians, non-musicians) and the within-subject factors emotion (happiness, pleasure, fear, and sadness) and morph type (Full, F0, and Tbr). A summary of all main effects and interactions is displayed in Table 1.

#### 3.3.1. P200

The P200 showed an interaction of emotion and morph type. Follow-up comparisons revealed parameter-specific modulations for pleasure and fear only (Figure 3): For pleasure, P200 amplitude was smaller in the F0 compared to the timbre and full conditions, which did not differ (F0 vs. Full: |*t*(77)| = 2.68, *p* = 0.009, *d* = 0.31 [0.08 0.53]; F0 vs. Tbr: |*t*(77)| = 2.62, *p* = 0.010, *d* = 0.30 [0.07 0.53]; Tbr vs. Full: |*t*(77)| = 0.19, *p* = 0.849, *d* = 0.02 [−0.20 0.25]). For fear, P200 amplitude was larger in the F0 compared to the timbre and full conditions, which again did not differ (F0 vs. Full: |*t*(77)| = 3.04, *p* = 0.003, *d* = 0.35 [0.12 0.58]; F0 vs. Tbr: |*t*(77)| = 2.21, *p* = 0.030, *d* = 0.25 [0.02 0.48]; Tbr vs. Full: |t(77)| = 0.97, *p* = 0.334, *d* = 0.11 [−0.11 0.33]). There were no main effects or interactions involving the factor group (Table 1).

#### 3.3.2. N400

The N400 showed the main effects of morph type and emotion and an interaction between both factors (Figure 3). Follow-up comparisons on the interaction revealed parameter-specific modulations for happiness and fear only: For happiness, the amplitude in the full condition was less negative than in the F0 and timbre condition, which differed only marginally (F0 vs. Full: |*t*(77)| = 4.27, *p* < 0.001, *d* = 0.49 [0.25 0.71]; Tbr vs. Full: |*t*(77)| = 5.44, *p* < 0.001, *d* = 0.62 [0.37 0.86]; F0 vs. Tbr: |*t*(77)| = 1.76, *p* = 0.081, *d* = 0.20 [−0.02 0.43]). For fear, the amplitude in the full condition tended to be more negative than in the F0 but not the timbre condition, which again did not differ (F0 vs. Full: |*t*(77)| = 2.24, *p* = 0.022, d = 0.27 [0.04 0.49]; Tbr vs. Full: |*t*(77)| = 1.31, *p* = 0.192, *d* = 0.15 [−0.08 0.37]; F0 vs. Tbr: |*t*(77)| = 1.12, *p* = 0.268, *d* = 0.13 [−0.10 0.35]). There were no main effects or interactions involving the factor group (Table 1).

#### 3.3.3. LPP

For the LPP, we observed the main effects of morph type and emotion, as well as the three-way interaction involving emotion, morph type, and group (Figure 4).

Follow-up analyses on non-musicians revealed a main effect of morph type only (*F*(2,76) = 4.39, *p* = 0.016, Ω_p_^2^ = 0.08), but no interaction of morph type and emotion (*F*(6,228) = 0.77, *p* = 0.592). Thus, across all emotions, the LPP amplitude in non-musicians was larger in the full compared to F0 and timbre conditions, which did not differ (F0 vs. Full: |*t*(38)| = 2.73, *p* = 0.010, *d* = 0.44 [0.11 0.77]; Tbr vs. Full: |*t*(38)| = 2.72, *p* = 0.010, *d* = 0.44 [0.11 0.77]; F0 vs. Tbr: |*t*(38)| = 0.10, *p* = 0.912, *d* = 0.02 [−0.30 0.30]).

In contrast, follow-up analyses on musicians revealed not only the main effects of emotion (*F*(3,114) = 4.55, *p* = 0.005, Ω_p_^2^ = 0.08) and morph type (*F*(2,76) = 4.05, *p* = 0.021, Ω_p_^2^ = 0.07) but also their interaction (*F*(6,228) = 3.15, *p* = 0.008, Ω_p_^2^ = 0.05). Thus, full, F0, and timbre modulations differed between emotions in musicians: For happiness, LPP amplitude was larger in the full condition than in the F0 condition, which was, in turn, larger than in the timbre condition (F0 vs. Full: |*t*(38)| = 2.42, *p* = 0.020, *d* = 0.39 [0.06 0.72]; Tbr vs. Full: |*t*(38)| = 4.33, *p* < 0.001, *d* = 0.70 [0.34 1.05]; F0 vs. Tbr: |*t*(38)| = 2.22, *p* = 0.032, *d* = 0.36 [0.03 0.69]). For pleasure, LPP amplitude was larger in the full condition compared with the timbre condition, while the F0 condition failed to differ from both (F0 vs. Full: |*t*(38)| = 0.60, *p* = 0.555, *d* = 0.10 [−0.22 0.41]; Tbr vs. Full: |*t*(38)| = 2.80, *p* = 0.008, *d* = 0.45 [0.12 0.79]; F0 vs. Tbr: |*t*(38)| = 1.44, *p* = 0.158, *d* = 0.23 [−0.09 0.55]). For fear and sadness, no differences between morph types were observed.

## 4. Discussion

In the present study, we explored ERPs of musicians and non-musicians to vocal emotions expressed through F0 only, timbre only, or both. Manipulation of F0 and timbre modulated the P200, the N400, and the LPP, indicating that acoustic parameter information is accessed early and affects the whole time course of emotional voice processing irrespective of musical skills. However, during later emotion processing (LPP), only musicians used this parameter information in an emotion-specific manner. Thus, musicality seems to affect later more controlled aspects of emotional processing.

The present findings partly replicate the exploratory pattern observed in a previous ERP study using a similar design [13]. For both experiments, the parameter-specific modulations of the P200 and N400 differed across emotions, suggesting that timbre may play a relatively more important role in the processing of pleasure compared to the processing of happiness, fear, and sadness. On a more detailed level, however, the present effects are smaller and slightly shifted in time. The prominent and early F0 vs. timbre effect observed for happiness in [13] was only marginally present in the N400, and the small effect of sadness was no longer detectable. These differences may be due to some key adjustments in the study design: First, we employed a different voice-morphing approach using emotional averages instead of neutral voices as the reference category. Both types of references are assumed to be emotionally uninformative and result in parameter-specific voice morphs of comparable emotional quality (for details, refer to [49]). Nevertheless, they differ with regard to their acoustic composition, which likely affected sensory-driven ERP components. Second, we adjusted the behavioral task. In [13], participants entered a response in every trial right after voice onset. In the present paradigm, participants were randomly prompted for a response in about 10% of trials after voice offset to reduce potential motor confounds. As a consequence, while listening to the voices, participants did not know whether they would have to make a response, potentially resulting in task-related differential top–down modulation of neural activity between these two studies. In short, the present listeners’ focus may have been less on conscious emotion recognition, which could have made the acoustic manipulation less impactful and reduced ERP effects. This idea is in line with evidence showing that the direction of listeners’ attention can modulate ERPs related to vocal emotional processing as early as 200 ms past voice onset [14,16,17].

The LPP showed different patterns for musicians and non-musicians. In non-musicians, its amplitude decreased when the emotion was expressed by either F0 or timbre only relative to the full condition. Thus, across all emotions, the LPP was affected when either acoustic parameter was rendered uninformative, indicating that the perceived emotional significance of the stimulus declined. Musicians, however, displayed an emotion-specific pattern. For the negative emotions of fear and sadness, the LPP was comparable across conditions, suggesting that musicians were able to compensate for a missing parameter. For happiness and pleasure, however, missing parameter information impeded emotion processing. Moreover, this impediment was greater when the timbre was informative while F0 information was absent and vice versa. These data align with behavioral evidence, implying a special reliance on F0 contour in musicians when compared with non-musicians [29]. At the same time, the present findings can be interpreted to suggest a more general flexibility and sensitivity toward emotional voice cues [21]. Perhaps this makes musicians more proficient in the processing of negative emotions when compared with non-musicians, as, indeed, such a point has been made previously for sadness [70]. Yet, future research is needed to test this possibility, as the present results could be driven by stimulus specificities.

This study reveals important new insight into the timing of neural processes that are shaped by musicality, suggesting effects on later more controlled aspects of emotional processing. Although there is consensus in the literature that the benefit of musicality is substantially based on auditory sensitivity [21,25,29], behavioral evidence alone is insufficient in revealing whether this is due to early, automatic representations of acoustic information or the result of later more controlled aspects of voice processing. In the present study, we found evidence for the latter, suggesting that behavioral musicality effects could result from a more efficient use of acoustic information for conscious decision making instead of bottom–up sensory sensitivity. Auditory processing in daily life rarely requires an explicit evaluation and categorization of expressed emotions, an aspect that frequently challenges the ecological validity of emotion recognition paradigms [71]. For musicians, however, explicit evaluation of acoustically expressed emotions usually forms part of their analytic work with music and may thus be specifically trained and/or leveraged. Against this backdrop, one may speculate that musicians’ proficiency in using subtle acoustic information for explicit emotion evaluation rather than bottom–up responses to the sensory input accounts for the behavioral benefit. Interestingly, a role for musicality in the explicit evaluation of music would explain the absence of musicality effects during implicit vocal emotional processing [45]. Accordingly, group differences in the present study may have been larger if we had prompted a behavioral response on every trial, a hypothesis that can be tested in the future.

The present absence of early ERP differences between musicians and non-musicians may be surprising as, indeed, they were variably identified in previous studies covering a broad range of auditory processes apart from those pursued here. For example, one study observed group differences in auditory brainstem potentials as early as 20 ms, implying that, at this early stage, musicians represent the complexity of an auditory signal better than non-musicians [35]. Milovanov and coworkers [72] found that children aged 10–12 years with more advanced musicality and pronunciation skills exhibited a more prominent mismatch negativity (MMN, 200–400 ms) compared to their less advanced peers in both musical and vowel stimuli in which the deviant was defined by stimulus duration. A more recent study [73] investigated 64 children aged 8–11 who were assigned to two groups according to whether or not they had music as a hobby. In this study, MMN amplitudes (measured from 150–250 ms) did not differ between musical and non-musical children when the paradigm used simple duration-defined deviants. By contrast, larger MMN amplitudes for musical children were observed in a multi-feature paradigm, in which deviants were defined by concomitant changes in duration, intensity, and location. The perhaps most relevant investigation for the present purpose pursued effects of musicality on early ERP components (P50, N100, and P200) depending on whether emotional prosody was presented with intelligible or unintelligible semantic content [43]. However, this small sample study (*n* = 14 per group) failed to identify differences between neutral and emotional prosody. Thus, evidence for early ERP differences between musicians and non-musicians in response to vocal emotions remains inconclusive.

Interestingly, however, Santoyo et al. [74] observed evidence in line with the present late musicality effects. These authors investigated the timbre-induced music-like perception of pitchless sounds in small groups of young adult musicians and non-musicians (*n* = 11 and 12, respectively) using a 64-channel EEG system similar to the present study. They found that group differences in the ERP started around 325 ms and continued until about 1200 ms after stimulus onset. Moreover, an analysis of induced neural oscillations in the same data also pointed to late differences between musicians and non-musicians, starting at about 650 ms. These findings of musicality effects on the processing of specific acoustic information are broadly consistent with the present data.

Overall, we wish to emphasize that the absence of an early (<500 ms) musicality effect in the present data does not exclude the possibility that such an effect exists [35,42,43]. Rather, different methods to analyze EEG brain activity in relationship to stimulus processing can be differentially (in)sensitive to different (and a priori unknown) processing aspects. In fact, there are examples of absent condition differences in early auditory ERPs that emerged when the same data were analyzed differently by, for example, computing induced neural oscillations [75] or conducting multivariate pattern analyses (MVPA) [76]. Here, we opted for conventional ERP analysis because of our interest in the different voice processing stages that have been pinned to different ERP components [13]. Even though reports that combine such different analysis methods with the same EEG datasets are rare, future research may benefit from such an approach and further enhance our understanding of neural differences related to musicality.

To the best of our knowledge, this is the first study using a voice manipulation technique to study individual differences in the electrophysiological processing of vocal emotions. This approach revealed novel insights into the time course of emotional F0 and timbre processing, as well as musicality-related differences in later ERP components. However, the degree to which the present findings generalize across different stimuli and tasks remains to be assessed in future research. Looking more broadly, the present paradigm illustrates the great potential of combining parameter-specific voice morphing with ERP measures, which could expand our understanding of the brain mechanisms underpinning auditory processing and their interindividual variations in vocal emotions and beyond.

## 5. Conclusions

In this ERP study, we presented manipulated voices that expressed emotional information through F0, timbre, or both. This acoustic manipulation had extensive effects on vocal emotional processing in the brain by modulating the P200, N400, and LPP. In later processing stages (LPP), this modulation interacted with musicality. In non-musicians, a missing parameter reduced the emotional significance of vocalizations as indexed by the LPP, irrespective of the parameter and the emotion. In musicians, however, parameter effects were more nuanced. For some emotions, a missing parameter was readily compensated, whereas for others, it dampened emotional responses. Moreover, this dampening was most pronounced with F0 absent, supporting a special link between musicality and F0-related processes. Together, these results show that, unlike non-musicians, musicians use parameter information in an emotion- and parameter-specific manner during late evaluative processing.

## Figures and Tables

**Figure 1 brainsci-13-01563-f001:**
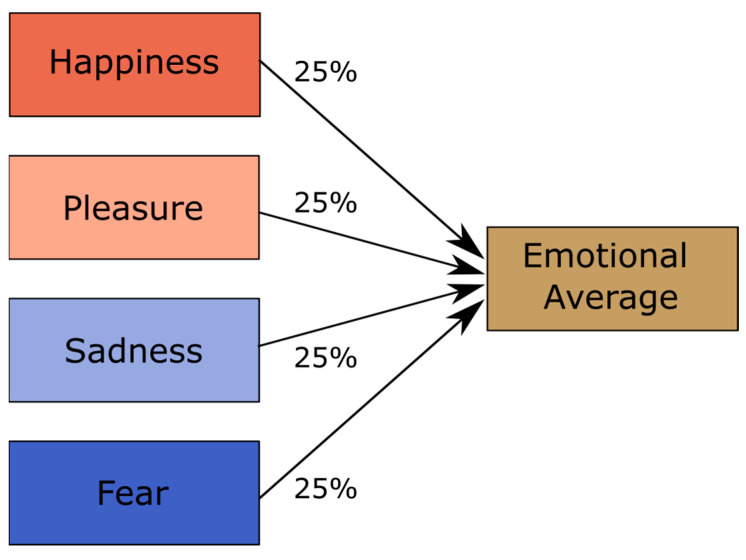
Illustration of the voice averaging process.

**Figure 2 brainsci-13-01563-f002:**
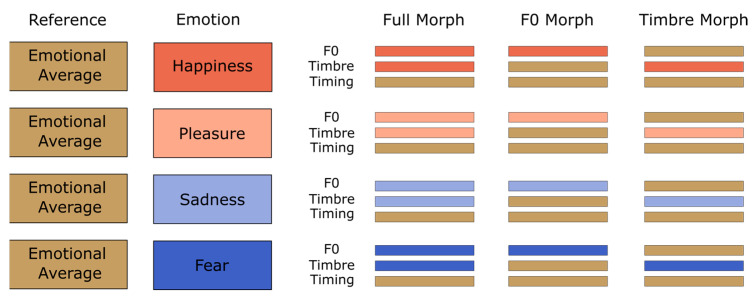
Morphing matrix for stimuli with averaged voices as reference.

**Figure 3 brainsci-13-01563-f003:**
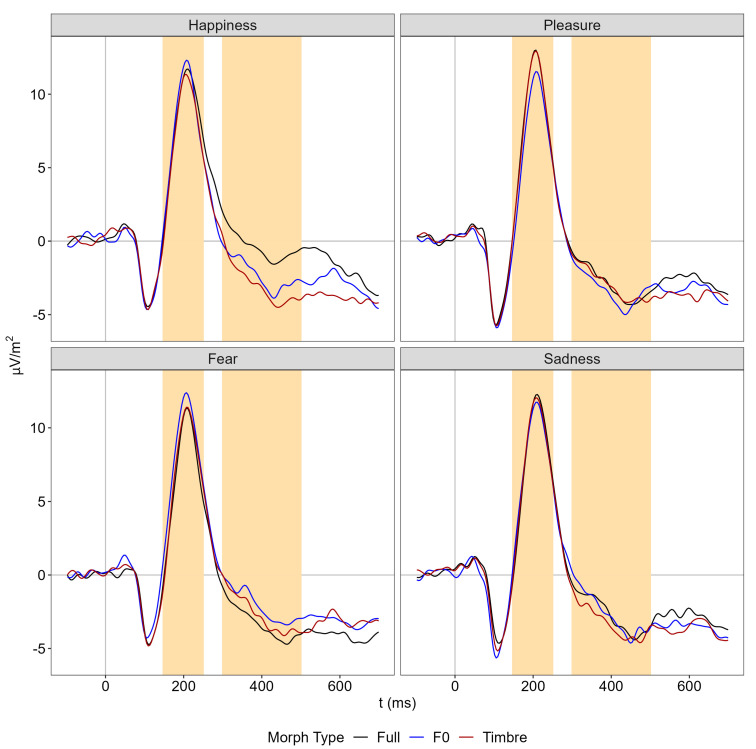
Fronto-central ERPs separately for emotion and morph type averaged across nine channels (details in Figure 4B). Yellow shaded areas illustrate the analysis window of the P200 (150 to 250 ms) and the N400 (300 to 500 ms).

**Figure 4 brainsci-13-01563-f004:**
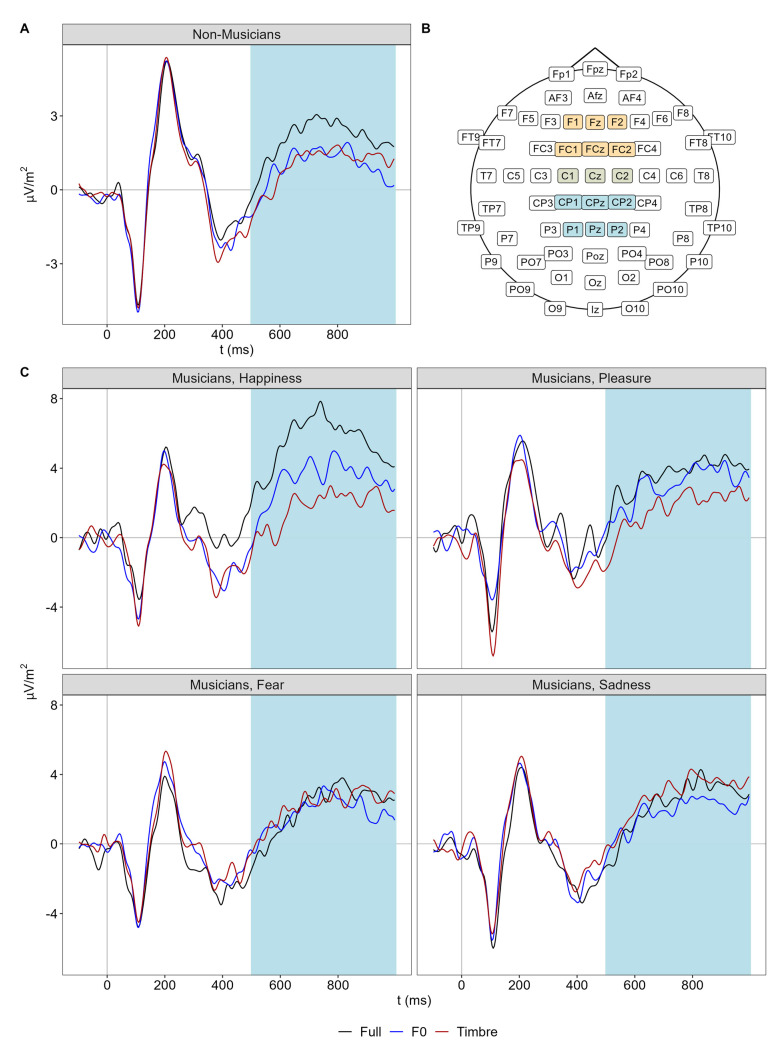
(**A**) Centro-parietal ERPs of non-musicians for each morph type separately. (**B**) EEG channel locations with the fronto-central cluster in yellow and the centro-parietal cluster in light blue. Note that both clusters overlap at C1, Cz, and C2, marked in light green. (**C**) Centro-parietal ERPs of musicians for each emotion and morph type separately. Light-blue shaded areas illustrate the analysis window of the LPP (500 to 1000 ms).

**Table 1 brainsci-13-01563-t001:** Results of the 2 × 4 × 3 mixed-effect ANOVAs on amplitudes of the P200, N400, and LPP.

		P200			N400			LPP		
	*df*1|2	*F*	*p*	Ω_p_^2^	*F*	*p*	Ω_p_^2^	*F*	*p*	Ω_p_^2^
Group	1|76	0.78	0.380	<0.01	0.04	0.848	<0.01	2.15	0.147	0.03
Emotion (Emo)	3|228	1.52	0.211	<0.01	**6.96**	**<0.001**	**0.084**	**5.24**	**0.002**	**0.07**
Morph Type (MType)	2|152	0.13	0.882	0.1	**3.80**	**0.024**	**0.048**	**8.18**	**<0.001**	**0.10**
Group × Emo	3|228	0.55	0.651	<0.01	0.51	0.679	0.017	1.71	0.166	0.02
Group × Mtype	2|152	1.04	0.356	<0.01	0.18	0.832	<0.01	0.23	0.791	<0.01
Emo × Mtype	6|456	**3.80**	**0.001**	**0.04**	**5.44**	**<0.001**	**0.067**	1.58	0.156	0.02
Group × Emo × Mtype	6|456	1.01	0.416	<0.01	0.83	0.548	0.1	**2.49**	**0.025**	**0.03**

Note: Outcome variables were mean amplitudes of the P200, the N400, and the LPP. Bold values highlight the significant effects. Effect sizes are displayed as partial omega squared (Ω_p_^2^), which is assumed to be less biased than eta squared [68,69].

## Data Availability

Appendix A, analysis scripts, and pre-processed data can be found on the associated OSF repository https://osf.io/2jt5h/ (27 September 2023).

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
