# Peer review of "Electrophysiological Correlates of Vocal Emotional Processing in Musicians and Non-Musicians"

_brainsci, 2023, doi:10.3390/brainsci13111563_

Round 1
Reviewer 1 Report
Comments and Suggestions for Authors
I found the article quite interesting. The authors conducted a comparison of event-related potentials (specifically P200 (150-250 ms) and N400 (300-500 ms) components and Late Positive Potential (500-1000 ms)) in response to acoustically manipulated voices, contrasting musicians and non-musicians. Although the study was well conducted, I believe that the discussion section would benefit from a deeper exploration of why there were no early ERP differences between musicians and non-musicians in the processing of vocal emotion. It would be valuable to include references to existing literature that supports this particular finding. Furthermore, I am particularly interested in gaining more insight into the differences in emotion-specific patterns within the LPP between musicians and non-musicians.
The subsequent neurophysiological study following the behavioral study is intriguing and could make a more significant contribution to the existing literature if the authors expanded on the theoretical rationale for their research, offered a more critical evaluation of the results, and discussed the potential implications for underlying mechanisms and distinctions between the two groups.
In addition, it is important to note that it is challenging to refer to the methods described in Nussbaum, C.; Schirmer, A.; Schweinberger, S.R. Musicality – Tuned to the Melody of Vocal Emotions. In press in 509 British Journal of Psychology 2023 during the review process, as this particular study is not accessible. This omission leaves readers without the means to understand the methods used. Addressing this issue would greatly improve the clarity and understandability of the paper.
I look forward to revisiting the article once the authors have addressed these issues.
I believe that this paper has considerable potential to make a meaningful contribution and I hope that my feedback will help to improve it.
Introduction
The literature often refers to the same set of studies, mainly from a single research group. However, this can be seen as a matter of the preferred style of the authors and how they choose to present their work.
As I read through the introduction, I felt the need for a broader framework that delved into the neural processing of vocal emotion in the brain, encompassing distinct stages with identifiable sub-processes associated with different event-related potentials (ERPs). It would be beneficial to elaborate on the expected irregularities in musicians compared to non-musicians in the context of vocal emotion recognition. I recommend including additional general references that examine differences in ERP responses to vocal emotion between musicians and non-musicians to support your hypothesis.
Participants
How and from where were the musicians and non-musicians recruited?
What was the reason for not excluding non-musicians with musical training?
It would be helpful not only to rely on Table S1 for this information, but also to have statistical evidence in the text that there are no significant differences in terms of age and education among the participants, and that there are observable differences in musicality between musicians and non-musicians.
Original audio recordings:
What is the same and what exactly has changed for this study?
Method Section general:
Merely referencing a previous publication from the same research group, especially when that publication is unavailable, is inadequate. The current article should stand on its own and be comprehensible without the need to consult prior publications.
Tables S1-S7
Please number Tables S1-S7 in the order in which they appear in the text.
Procedure:
It is not clear which data from the previous online study were used in this study. The methods relevant to this research should be outlined in the methods section and analysed in the results section. As mentioned above, it should be ensured that the participants do not differ in terms of education and demographics, and that a distinction is made between musicians and non-musicians. The data collection procedure for this study should be clearly presented, and the relevant results should be discussed within the study, rather than simply extracted from the supplemental material.
Line 196-197: Which were the response options?
Line 203: Was determined whether they were right or left-handed?
Statistical analysis
The statistical analysis should be described in a separate section. Please be more specific on the designs of the statistical test (dependant and independent variables, exact procedure used, correction for multiple testing).
Results
As mentioned above, Tables S1-S7 should be numbered correctly.
The Chi-square test should be correctly reported in the text and not just in the table.
Discussion
I'm curious about the reasons for the lack of early ERP differences between musicians and non-musicians in their processing of vocal emotion. It would be beneficial to include references to previous or relevant existing research that supports this observation or/and clearly specify limitations of this study. In addition, I am particularly interested in gaining a deeper understanding of the differences in emotion-specific patterns within the LPP between musicians and non-musicians.
Please include the novel aspects of this study and the significance it holds for future research and potential practical applications.
Reviewer 2 Report
Comments and Suggestions for Authors
This paper deals with interesting contents, especially the use of ERPs in relation to emotional voicing is quite relevant. The paper is well-written and the methodology seems to be sound. The experimental design is also quite original, but the used argumentation is not always at high academic standards. There are also some problems with the readability and understandability of the paper, due to the use of technical jargon which is not always sufficiently explained. Some major and focal concepts of the paper are also not very well described. The theoretical background of the paper is also somewhat limited, and several claims are introduced without critical discussion or strong motivation. In order to improve the overall quality of the paper, I list below some general remarks and detailed comments.
General remarks
· The English language is fluent.
· The explanation of the study design is original but is not sufficiently explained.
· The methodology in general must also be explained better. The description must be self-sufficient rather than referring merely to some referencewhere the methods are described. This can be at times a difficult exercise, but a short intuitive description of the what and how of some methodological approach can help a lot to improve the readability and understandability of the paper?
· Explain better how pitch and timbre are modulated.
· What is exactly meant with pitch contour and its relation with F0. What is F0 contour? Explain clearly as this is a focal concept in the paper.
· Explain better how emotional pitch cues are related to F0.
· Explain better what is meant with parameter-specific ERP modulation. Provide a short intuitive description.
· Explain better the mechanisms of emotional averages. This seems to be a very interesting concept.
· The tables must be explained more in detail. How to interpret the data? PIease provide more details to avoid too much searching efforts by the reader.
· Omega squared is not frequently used and is not common knowledge for a mean readership. Explain very shortly that it is a measure of effect size and why it is used instead of eta square.
Detailed comments
· Page 2, line 51: please explain shortly and in intuitive terms what is meant with parameter-specific voice morphing, especially at first appearance of the term. The term is also very focal in the paper. It is also not immediately clear how voices can express emotions by merely changing the F0 and timbre. Changing solely F0 seems to be quite reductionistic, and timbre is also a very general category. How is the timbre modulated or changed?
· Page 2, line 53: it is not clear how F0 can be described as the perceived pitch contour. It seems that F0 is just the fundamental frequency, not the contour. It is meant here that a succession of fundamental frequencies builds up a pitch contour? This could make sense, but not the wording as it is now. This is quite confusing.
· Page 2, line 72: same remark.
· Page 2, line 77: what is meant with emotional pitch cues? This seems to be a very crude generalization as if heightening or decreasing of the fundamental frequency leads automatically to a change in evoked emotional reaction. This needs more critical discussion and motivation.
· Page 2, line 82: what is meant with an implicit listening task? Explain shortly.
· Page 2, line 89: explain shortly how sensitivity to F0 and timbre cues is measured. Which timbre cues?
· Page 3, line 102: this is potentially an important statement, but it is not clear how the reader must interpret the word “modulation”. This can be interpreted as if the modulation is done by an external agent (the common definition). But this seems not to be the case here. Should it be understood as if the listener is modulating his/her own components? In that case, explain this better to avoid confusion.
· Page 3, line 108: the concept of behavioral online study is introduced without explaining which kind of behavior. This should either be explained very shortly or a reference to a later place in the paper should be inserted (something as: see below).
· Page 3, line 133: what does the abbreviation OSF stand for? Please provide the full wording.
· Page 4, line 148: the concept of emotional average seems to be quite eye-opening, but it is not explained how this averaging is done. Please provide a clear description of the process.
· Page 4, line 157: please explain a little more what is meant with the concept of aperiodicity.
· Page 5, line 230: the “spherical pine procedure” is an established statistical technique, but is quite technical, and may not be known very well by the common readership. Explain very shortly what it means and what it is used for.
· Page 5, line 233: same remark for “current source density”
· Page 6, line 240; P200 [150, 250]. Do the numbers between brackets refer to the boundaries of the analysis window? Please explain at first introduction of the data, either in the main text or in the table caption.
· Page 6, line 262: please explain better to what extent the data are behavioral. It seems that the subject must do something, a kind of manifest behavior. Please explain more intuitively so as to be very clear.
· Page 6, table 1: explain the table a little more in detail. How to interpret the data. You use also omega squared and not eta squared. As this measure is less well known please explain shortly that is refers to effect size, and explain also why you use omega instead of eta.
Round 2
Reviewer 1 Report
Comments and Suggestions for Authors
Thank you for your rework.
Author Response
Thank you very much for your time and your valuable feedback!